# Clinical Characteristics of Patients with Post-Tuberculosis Bronchiectasis: Findings from the KMBARC Registry

**DOI:** 10.3390/jcm10194542

**Published:** 2021-09-30

**Authors:** Hayoung Choi, Hyun Lee, Seung Won Ra, Hyun Kuk Kim, Jae Seung Lee, Soo-Jung Um, Sang-Heon Kim, Yeon-Mok Oh, Yong-Soo Kwon

**Affiliations:** 1Division of Pulmonary, Allergy, and Critical Care Medicine, Department of Internal Medicine, Hallym University Kangnam Sacred Heart Hospital, Hallym University College of Medicine, Seoul 07441, Korea; hychoimd@gmail.com; 2Division of Pulmonary Medicine and Allergy, Department of Internal Medicine, Hanyang University College of Medicine, Seoul 04763, Korea; namuhanayeyo@hanyang.ac.kr (H.L.); sangheonkim@hanyang.ac.kr (S.-H.K.); 3Division of Pulmonary Medicine, Department of Internal Medicine, Ulsan University Hospital, College of Medicine, University of Ulsan, Ulsan 44033, Korea; docra@docra.pe.kr; 4Division of Pulmonary and Critical Care Medicine, Department of Internal Medicine, Inje University Haeundae Paik Hospital, Busan 48108, Korea; khkmd1205@hanmail.net; 5Department of Pulmonary and Critical Care Medicine, Asan Medical Center, University of Ulsan College of Medicine, Seoul 05505, Korea; jsdoc1186@daum.net (J.S.L.); yeonmok.oh@gmail.com (Y.-M.O.); 6Department of Internal Medicine, Dong-a University Hospital, Busan 49201, Korea; sjum@dau.ac.kr; 7Department of Internal Medicine, Chonnam National University Medical School, Chonnam National University Hospital, Gwangju 61469, Korea

**Keywords:** bronchiectasis, tuberculosis, treatment

## Abstract

The clinical characteristics of patients with post-tuberculosis (TB) bronchiectasis have not been well evaluated. We enrolled 598 patients with bronchiectasis who participated in the Korean prospective bronchiectasis registry and compared the characteristics of post-TB bronchiectasis (19.7%) with post-infectious (19.6%), idiopathic (40.8%), and other (19.9%) bronchiectasis. The patients with post-TB bronchiectasis had a lower body mass index, higher rate of chronic obstructive pulmonary disease, and lower rate of asthma than those in the other groups. The patients with post-TB bronchiectasis had more upper lobe involvement, more severe radiological extent, and worse lung function than those in the other groups. Long-acting muscarinic antagonist/long-acting ß agonist use and mucolytics were more commonly used in the patients with post-TB bronchiectasis than those in the other groups, while inhaled corticosteroid/long-acting ß agonist was less commonly used. There were no significant intergroup differences in bronchiectasis severity scores except for FACED, the number of exacerbations, and quality of life. Post-TB bronchiectasis is characterised by reduced lung function and higher rates of mucolytic use when compared with other bronchiectasis; thus, adequate bronchodilator use and airway clearance techniques may alleviate symptom burden in this population.

## 1. Introduction

The diagnosis of non-cystic fibrosis bronchiectasis (hereafter referred to as bronchiectasis) is increasing worldwide [1,2]. The increased prevalence of bronchiectasis has led to the establishment of several prospective multicentre bronchiectasis registries [3,4,5]. Research from the international registries revealed that the aetiologies of bronchiectasis varied according to region, and tuberculosis (TB) was most common in Asian countries, including India and South Korea [6,7]. Therefore, elucidating specific clinical characteristics of patients with post-TB bronchiectasis is essential to successfully manage Asian patients with bronchiectasis.

Most studies on post-TB bronchiectasis have been performed on patients who survived after TB treatment. Bronchiectasis is a form of TB-associated lung damage that results from the destruction of elastic and muscular components of the bronchial walls [8]. Patients with post-TB bronchiectasis typically show airflow obstruction in addition to recurrent episodes of purulent sputum production and haemoptysis [9,10]. Although two studies assessed the performance of bronchiectasis severity scoring systems in post-TB bronchiectasis [11,12], there is limited information on the clinical characteristics of post-TB bronchiectasis in the context of a bronchiectasis cohort.

This study aimed to assess the clinical characteristics, respiratory treatment, exacerbations, and quality of life of patients with post-TB bronchiectasis compared with those with other bronchiectasis using a prospective multicentre bronchiectasis cohort.

## 2. Materials and Methods

### 2.1. Study Design and Participants

This study included participants enrolled in the Korean Multicentre Bronchiectasis Audit and Research Collaboration (KMBARC) registry between August 2018 and December 2019 [3]. The study population comprised adult patients (age ≥ 18 years) with stable bronchiectasis confirmed with chest computed tomography (CT). Exclusion criteria were as follows: (1) bronchiectasis due to cystic fibrosis; (2) traction bronchiectasis associated with interstitial lung disease; (3) patients being actively treated for pneumonia, pulmonary TB, or non-tuberculous mycobacteria (NTM) infection; (4) patients who were unable or unwilling to provide informed consent; and (5) pregnant patients. The detailed registry protocol and baseline characteristics of the patients were described previously [3,6].

This study protocol was approved by the institutional review board of all institutions participating in the KMBARC registry, including Hallym University Kangnam Sacred Heart Hospital (application no. 2018-07-16), and written informed consent was obtained from all patients.

### 2.2. Bronchiectasis Aetiology and Definition of Post- TB Bronchiectasis

The attending physicians determined the aetiology of bronchiectasis based on the test results and questionnaires according to published international guidelines [13,14]. Post-TB bronchiectasis was assigned when history or clinical evidence of TB was evident, and radiological findings of bronchiectasis were suggestive of TB, including scarring of the upper lobes, granuloma, and cavitation.

The post-infectious aetiology was defined as bronchiectasis caused by a prior infection other than TB. Idiopathic bronchiectasis was defined when no specific aetiology was determined after diagnostic workup according to the guidelines [13,14].

### 2.3. Assessments

The KMBARC registry collected baseline data at baseline (recruitment) and follow-up visits once a year. The collected data included demographics, comorbidities, laboratory test results (including microbiology), radiology, pulmonary function test results, and treatment [3].

Dyspnoea was assessed using the modified Medical Council Dyspnoea (mMRC) scale. Pulmonary function testing was performed according to the European Respiratory Society and American Thoracic Society technical standards and percentage of predicted forced expiratory volume in 1 s (FEV_1_) calculated using reference values for Korean patients [15,16]. Airflow obstruction was defined as an FEV_1_/forced vital capacity (FVC) ratio less than 0.7 [17]. Regarding radiologic findings, the number of involved lobes and the presence of bronchiectasis at each lobe were assessed using chest CT. Additionally, a modified Reiff score was used to evaluate radiologic severity [18]. Data on respiratory medications (inhalers, mucolytics, and long-term antibiotics), home oxygen, and physiotherapy were obtained.

The severity of bronchiectasis was determined using the bronchiectasis severity index (BSI), FACED, and E-FACED [19,20]. A consensus definition was used to determine bronchiectasis exacerbation and defined as deterioration in three of the following symptoms for at least 48 h: (1) cough; (2) sputum volume increase and/or consistent change; (3) sputum purulence; (4) dyspnoea and/or exercise intolerance; (5) fatigue and/or malaise; or (6) haemoptysis [21]. This study also used the following measurement tools related to health-related quality of life: the validated Korean version of the Bronchiectasis Health Questionnaire (BHQ) for quality of life, Patient Health Questionnaire 9 for depression, and Fatigue Severity Score for fatigue [3,22].

### 2.4. Statistical Analyses

Continuous data are presented as medians with interquartile ranges (IQRs), and results were compared using the Kruskal–Wallis test. Categorical data are presented as numbers (%), and results were compared using Pearson’s chi-squared test or Fisher’s exact test, as appropriate. Statistical significance was set at *p* < 0.05. To account for multiple comparisons, post hoc Bonferroni correction was applied if the Kruskal–Wallis test or Pearson’s chi-squared test indicated significance, in which a p value of 0.05 corresponds to 0.008 (0.05/6). All statistical analyses were performed using Stata (Release 16; StataCorp LP, College Station, TX, USA), and graphs were compiled with the use of GraphPad Prism version 9.0.2 (GraphPad Software, San Diego, CA, USA).

## 3. Results

### 3.1. Baseline Characteristics

A total of 598 patients were enrolled in this study, comprising 118 (19.7%) patients with post-TB, 117 (19.6%) with post-infectious, 244 (40.8%) with idiopathic, and 119 (19.9%) with other bronchiectasis. Common aetiologies among the other bronchiectasis included asthma (*n* = 32/119), NTM pulmonary disease (*n* = 24/119), chronic obstructive pulmonary disease (COPD) (*n* = 21/119), and rheumatoid arthritis (*n* = 15/119). The median patient age was 66 years (IQR, 60–72 years) and 56% were females. The patients with post-TB bronchiectasis showed a lower body mass index than the other three bronchiectasis groups (*p* = 0.002). There were no significant intergroup differences in smoking history, sputum volume, or rate of having severe dyspnoea ≥ 2 on the mMRC scale. Regarding pulmonary comorbidities, the patients with post-TB bronchiectasis had a significantly higher rate of COPD (*p* < 0.001) and lower rate of asthma (*p* < 0.001) than those in the other groups. Regarding extrapulmonary comorbidities, the patients with post-TB bronchiectasis had lower rates of rheumatoid arthritis (*p* = 0.008) and depression (*p* = 0.024) than those in the other groups (Table 1).

### 3.2. Radiologic, Pulmonary Function, Microbiologic, and Laboratory Test Results and Respiratory Treatment 

The patients with post-TB bronchiectasis had significantly more involvement in the right upper lobes (*p* < 0.001) and upper divisions of the left upper lobes (*p* < 0.001) than those in the other groups. In contrast, the patients with post-TB bronchiectasis had less involvement in the right lower lobes (*p* = 0.047) and left lower lobes (*p* = 0.010) than those in the other groups. The patients with post-TB and post-infectious bronchiectasis had higher modified Reiff scores than those with other bronchiectasis (*p* = 0.002). Regarding pulmonary function tests, the patients with post-TB bronchiectasis had a lower FEV1/FVC ratio (*p* = 0.025) than those in the other groups; the rate of patients with FEV1/FVC < 0.7 tended to be higher in the patients with post-TB and post-infectious bronchiectasis than those with idiopathic and other bronchiectasis, although the difference was not statistically significant (*p* = 0.075). Furthermore, the patients with post-TB bronchiectasis had worse lung functions, including FEV1, %predicted (*p* < 0.001) and FVC, %predicted (*p* = 0.027) than the patients in the other groups. Concerning microbiologic test results, the patients with post-TB and other bronchiectasis showed higher rates of NTM than those with post-infectious and idiopathic bronchiectasis (Table 2). 

The patients with post-TB bronchiectasis used long-acting muscarinic antagonist (LAMA)/long-acting ß agonist (LABA) more frequently than those with other bronchiectasis (*p* < 0.001); however, inhaled corticosteroid (ICS)/LABA was much less frequently used in the patients with post-TB bronchiectasis than in those with other bronchiectasis (*p* = 0.001). Mucolytics were prescribed significantly more frequently in the patients with post-TB and post-infectious bronchiectasis than in those with idiopathic and other bronchiectasis (*p* < 0.001) (Figure 1).

### 3.3. Bronchiectasis Severity, Exacerbations, and Quality of Life

Among the disease severity scores, only FACED was higher among the patients with post-TB bronchiectasis compared with those in the other groups (*p* = 0.042). Additionally, there was no significant intergroup differences in exacerbations, history of haemoptysis requiring hospitalisation, and quality of life between the four patient groups (Table 3).

## 4. Discussion

This Korean prospective bronchiectasis registry study revealed that approximately one-fifth of all the patients with bronchiectasis had post-TB bronchiectasis. The patients with post-TB bronchiectasis had more severe radiological extent and worse lung function than those with other bronchiectasis, which was reflected in the more frequent use of inhaled bronchodilators among the patients with post-TB bronchiectasis than those in the other groups. However, there were no significant intergroup differences in bronchiectasis severity scores (except for FACED), number of exacerbations, or quality of life between the four patient groups.

In this study, approximately two-thirds of the patients with post-TB bronchiectasis had airflow limitation, and these patients had a significantly lower FEV_1_ compared with those with idiopathic and other bronchiectasis. Considering the importance of identifying treatable traits in the patients with bronchiectasis to improve treatment outcome [23], the high proportion of subjects with an advanced degree of airflow limitations among the patients with post-TB bronchiectasis indicates that bronchodilators might be an attractive treatment strategy to improve lung function in these patients. Previous studies showing that bronchodilators are effective to improve lung function in bronchiectasis patients with airflow limitations further support our view [24,25]. However, as only a few retrospective studies have evaluated the effect of bronchodilators on lung function change in bronchiectasis patients [24,25], future studies with comprehensive study outcomes, including not only lung function but also quality of life and acute exacerbations, are needed. In addition, as smoking can enhance the development of airflow limitations in patients with post-TB bronchiectasis [26], strategies for enhancing smoking cessation would be important for patients with post-TB bronchiectasis. 

The patients with post-TB bronchiectasis had more severe radiologic extent, measured using the modified Reiff score, than the patients with other bronchiectasis. In addition to the chest CT findings, a much higher rate of mucolytic use may suggest a higher symptom burden in the post-TB bronchiectasis group. Consequently, airway clearance techniques need to be emphasized more in patients with post-TB bronchiectasis. Such airway clearance techniques and chest physiotherapy may ultimately improve long-term outcomes such as the frequency and severity of exacerbations, hospitalisations, and quality of life in this population [27,28]. Although our study results showed no intergroup differences in exacerbations or bronchiectasis severity scores, this might be due to its cross-sectional design, which measured outcomes at one-time point. Future longitudinal study results using KMBARC follow-up data will prove whether post-TB aetiology is related to any difference in long-term outcomes among patients with bronchiectasis.

Another notable finding was that the patients with post-TB bronchiectasis showed higher NTM colonization than those with post-infectious and idiopathic bronchiectasis. TB has been suggested as a risk factor for incident NTM infection [29,30], as was shown in our study results. Considering that long-term macrolide monotherapy increases the risk of macrolide-resistant NTM pulmonary disease, a difficult-to-treat disease, it is imperative to collect sputum samples for NTM culture, especially in patients with post-TB bronchiectasis, before starting long-term macrolide treatment [30,31]. Regarding ICS use, the patients with post-TB, post-infectious, and idiopathic bronchiectasis were less likely to use ICS than those with other bronchiectasis where asthma or COPD comprised a large proportion. As prolonged use of ICS can lead to mycobacterial infection, a limited use of ICS is recommended in patients with bronchiectasis [13,14]. Thus, our study results suggest that most of our study population adheres to the current recommendation [13,14].

The major strength of our study is that it is the first to use a prospective bronchiectasis registry for patients with post-TB bronchiectasis. However, this study has limitations that should be acknowledged. First, this study only analysed baseline data from the KMBARC registry. Given the nature of a cross-sectional study, future research using longitudinal KMBARC registry data is warranted to evaluate the clinical characteristics of patients with post-TB bronchiectasis, including exacerbations, lung function decline, and long-term prognosis. Second, this study was performed in Korea, which limits the generalisation of our findings to other institutions in geographically different regions. Third, physicians’ judgments regarding the aetiology of bronchiectasis could be subjective. Although all the patients enrolled in the KMBARC were evaluated using the same clinical report form (same tests and questionnaire), the intensity of the diagnostic workup might differ between institutions and physicians. Since a diagnostic bundle for bronchiectasis in South Korea has been developed (unpublished data), this issue should be addressed in future studies.

## 5. Conclusions

Our study showed that approximately one-fifth of all the patients with bronchiectasis had post-TB bronchiectasis. The patients with post-TB bronchiectasis had more severe radiological extent, reduced lung function, and more frequent use of inhaled bronchodilators and mucolytics compared with the patients with other aetiologies. There were no significant intergroup differences in the bronchiectasis severity scores except for FACED, the number of exacerbations, and quality of life. Adequate bronchodilator use and airway clearance techniques may alleviate the symptom burden in patients with post-TB bronchiectasis. 

## Figures and Tables

**Figure 1 jcm-10-04542-f001:**
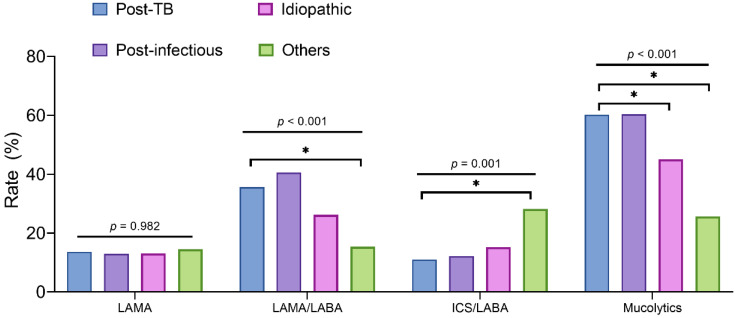
Comparison of respiratory medication use in patients with bronchiectasis according to their aetiologies. * Indicates statistical significance for the comparison of the post-TB group with each other. To account for multiple comparisons, post hoc Bonferroni correction was applied, in which a *p* value of 0.05 corresponded to 0.008 (0.05/6). TB, tuberculosis; LAMA, long-acting muscarinic antagonist; LABA, long-acting ß agonist; ICS, inhaled corticosteroid; OCS, oral corticosteroid; anti, oral antibiotics.

**Table 1 jcm-10-04542-t001:** Baseline characteristics of the study population.

	Post-TB (*n* = 118)	Post-Infectious (*n* = 117)	Idiopathic (*n* = 244)	Others (*n* = 119)	*p* Value
Age, years	66 (60–72)	67 (60–71)	65 (60–71)	66 (60–73)	0.408
Males	62 (52.5)	50 (42.7)	100 (41.0)	52 (43.7)	0.215
BMI, kg/m^2^	22.1 (19.5–24.6)	22.6 (20.4–24.6)	23.2 (21.0–25.8) *	23.2 (20.7–25.1)	0.002
Smoking history					0.349
Never-smoker	68 (57.6)	77 (65.8)	162 (66.4)	80 (67.2)	
Current- or ex-smoker	50 (42.4)	40 (34.2)	82 (33.6)	39 (32.8)	
Sputum volume, mL	10 (5–25)	20 (5–50)	11 (5–32)	10 (5–25)	0.701
mMRC dyspnoea scale					0.107
<2	83 (70.3)	91 (77.8)	199 (81.6)	95 (79.8)	
≥2	35 (29.7)	26 (22.2)	45 (18.4)	24 (20.2)	
Comorbidities					
Pulmonary comorbidities					
COPD	53 (44.9)	59 (50.4)	69 (28.3) *	45 (37.8)	<0.001
Asthma	17 (14.4)	31 (26.5)	37 (15.2)	49 (41.2) *	<0.001
Cardiovascular diseases					
Myocardial infarction	2 (1.7)	2 (1.7)	2 (0.8)	2 (1.7)	0.759
Angina	4 (3.4)	3 (2.6)	6 (2.5)	6 (5.0)	0.592
Stroke or TIA	2 (1.7)	2 (1.7)	3 (1.2)	4 (3.4)	0.524
CHF	2 (1.7)	5 (4.3)	4 (1.6)	2 (1.7)	0.450
Rheumatoid arthritis	5 (4.2)	6 (5.1)	10 (4.1)	16 (13.6)	0.008
Liver cirrhosis	4 (3.4)	3 (2.6)	1 (0.4)	0	0.024
Chronic kidney disease	2 (2.6)	3 (2.6)	4 (1.7)	2 (1.7)	0.891
Malignancy	16 (13.7)	13 (11.1)	16 (6.6)	9 (7.8)	0.127
Diabetes mellitus	13 (11.1)	11 (9.4)	34 (13.9)	15 (12.6)	0.640
Osteoporosis	13 (11.1)	22 (19.0)	22 (9.0)	13 (10.9)	0.052
Depression	0	5 (4.3)	11 (4.5)	4 (6.7)	0.024

Data are presented as number (percentage) or median (interquartile range). * Indicates statistical significance for the comparison of the post-TB group with each other. To account for multiple comparisons, post hoc Bonferroni correction was applied, in which a *p* value of 0.05 corresponded to 0.008 (0.05/6). TB, tuberculosis; BMI, body mass index; mMRC, modified Medical Research Council; COPD, chronic obstructive pulmonary disease; TIA, transient ischemic attack; CHF, congestive heart failure.

**Table 2 jcm-10-04542-t002:** Radiologic, pulmonary function, microbiologic, and laboratory test results in patients with bronchiectasis according to their aetiologies.

	Post-TB (*n* = 118)	Post-Infectious (*n* = 117)	Idiopathic (*n* = 244)	Others (*n* = 119)	*p* Value
Radiology					
No of involved lobes	3 (2–5)	3 (2–5)	3 (2–4)	3 (2–4)	0.052
Involved lobe					
RUL	74 (63.3)	49 (41.9) *	88 (37.0) *	36 (32.7) *	<0.001
RML	66 (56.4)	76 (65.0)	144 (60.5)	75 (68.2)	0.262
RLL	59 (50.4)	80 (68.4)	143 (60.1)	64 (58.2)	0.047
LUL upper division	63 (53.9)	46 (39.3)	79 (33.2) *	26 (23.6) *	<0.001
LUL lingular division	58 (49.6)	65 (55.6)	137 (57.6)	51 (46.4)	0.191
LLL	79 (67.5)	100 (85.5)	180 (75.6)	78 (70.9)	0.010
Modified Reiff score	6.4 ± 3.8	6.8 ± 4.2	6.0 ± 4.2	4.9 ± 3.3 *	0.002
Pulmonary function					
FVC, L	2.3 (1.9–3.1)	2.5 (2.0–3.0)	2.5 (2.0–3.0)	2.5 (2.0–3.2)	0.887
FVC, % predicted	70.1 (54.4–84.4)	72.6 (60.5–79.8)	74.4 (64.1–85.6) *	74.7 (63.2–82.9)	0.027
FEV_1_, L	1.5 (1.1–1.9)	1.5 (1.1–1.9)	1.7 (1.3–2.1)	1.7 (1.2–2.1)	0.163
FEV_1_, % predicted	57.6 (43.0–74.2)	58.7 (46.5–71.3)	65.8 (53.2–81.2) *	64.4 (51.6–77.6)	0.001
FEV_1_/FVC ratio	0.6 (0.5–0.7)	0.6 (0.5–0.7)	0.7 (0.6–0.8)	0.7 (0.5–0.7)	0.025
FEV_1_/FVC < 0.7	71 (65.7)	71 (68.9)	119 (55.1)	64 (61.0)	0.075
Microbiology					
NTM	16 (13.6)	9 (7.7)	15 (6.2)	22 (18.6)	0.001
*Pseudomonas aeruginosa*	23 (19.5)	23 (19.7)	50 (20.5)	10 (8.4)	0.030
*Haemophilus influenzae*	2 (1.7)	4 (3.4)	2 (0.8)	1 (0.8)	0.243
*Staphylococcus aureus*	0	1 (0.9)	2 (0.8)	1 (0.8)	0.921
*Moraxella catarrhalis*	0	0	2 (0.8)	1 (0.8)	1.000
*Enterobacteriaceae*	5 (4.2)	4 (3.4)	13 (5.3)	1 (0.8)	0.192
Laboratory					
WBC	6700 (5300–8300)	7200 (6100–9100)	7100 (5800–9000)	6500 (5800–9000)	0.149
Haemoglobin	13.3 (12.0–14.0)	13.3 (12.1–14.2)	12.9 (12.3–14.2)	13.4 (12.3–14.2)	0.689
ESR	23 (13–49)	29 (15–39)	26 (14–47)	25 (14–56)	0.925
CRP	0.6 (0.2–2.1)	0.4 (0.2–1.5)	0.4 (0.1–1.5)	0.5 (0.2–1.3)	0.274

Data are presented as number (percentage) or median (interquartile range), except for modified Reiff score, which is presented as mean ± standard deviation. * Indicates statistical significance for the comparison of the post-TB group with each other. To account for multiple comparisons, post hoc Bonferroni correction was applied, in which a *p* value of 0.05 corresponded to 0.008 (0.05/6). TB, tuberculosis; RUL, right upper lobe; RML, right middle lobe; RLL, right lower lobe; LUL, left upper lobe; LLL, left lower lobe; FVC, forced vital capacity; FEV_1_, forced expiratory volume in 1 s; NTM, non-tuberculous mycobacteria; WBC, white blood cell; ESR, erythrocyte sedimentation rate; CRP, C-reactive protein.

**Table 3 jcm-10-04542-t003:** Comparison of bronchiectasis severity, exacerbations, and quality of life of patients with bronchiectasis according to their aetiologies.

	Post-TB (*n* = 118)	Post-Infectious (*n* = 117)	Idiopathic (*n* = 244)	Others (*n* = 119)	*p* Value
Disease severity					
BSI	7 (5–9)	6 (5–9)	6 (4–9)	5 (4–8)	0.082
FACED	2.4 ± 1.6	2.2 ± 1.7	1.9 ± 1.6	1.9 ± 1.4	0.042
E-FACED	3 (1–4)	3 (1–3)	2 (1–3)	2 (1–3)	0.101
Exacerbations					
Any exacerbation	1 (0–3)	1 (0–2)	1 (0–2)	0 (0–2)	0.892
Requiring admission	22 (18.6)	22 (18.8)	45 (18.4)	20 (16.8)	0.976
Requiring ER visits	7 (5.9)	9 (7.7)	21 (8.6)	8 (6.7)	0.812
Severe exacerbation ≥ 2/year	6 (5.1)	8 (6.8)	12 (4.9)	6 (5.0)	0.887
Haemoptysis requiring admission	7 (6.1)	7 (6.2)	16 (7.0)	2 (2.4)	0.501
Bronchial artery embolization	3 (2.6)	1 (0.9)	3 (1.3)	0	0.422
Quality of life					
BHQ	53 (49–58)	50 (45–56)	53 (46–59)	53 (45–59)	0.111
PHQ-9	3 (1–9)	3 (1–9)	3 (1–8)	4 (2–10)	0.495
FSS	19 (12–33)	18 (12–32)	19 (12–32)	23 (14–37)	0.219

Data are presented as number (percentage) or median (interquartile range), except for FACED, which is presented as mean ± standard deviation. BSI, bronchiectasis severity index; ER, emergency room; BHQ, Bronchiectasis Health Questionnaire; PHQ-9, Patient Health Questionnaire 9; FSS, Fatigue Severity Score.

## Data Availability

The data presented in this study are available upon reasonable request from the corresponding author.

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
