# Peer review of "Clinical Characteristics of Patients with Post-Tuberculosis Bronchiectasis: Findings from the KMBARC Registry"

_jcm, 2021, doi:10.3390/jcm10194542_

Round 1
Reviewer 1 Report
The authors attempted to characterize patients with NCFBE and prior diagnosis of TB (as based solely on patients' reports). These patients seem to be sicker than NCFBE patients with no previous diagnosis of TB, and this is clearly and fluently described.
However, the major flaw of this study is that the actual association of prior TB with the severity of the present diagnosis of bronchiectasis was not proved, and indeed when comparing the patients with physician-diagnosis of post-TB-bronchiectasis to patients with prior TB but with other suspected etiology of bronchiectasis- no clinically significant difference was found.
So in conclusion, this study basically shows that in this cohort (and probably in the entire population), survivors of TB are in worse shape than people who hadn't suffered from TB, arguably regardless of the diagnosis of bronchiectasis (or at least not worse than bronchiectasis patients with other etiologies).
Therefore I fail to see the novelty or clinical significance of this data. Perhaps comparison of patients with physician diagnosed post-TB-bronchiectasis with patients with bronchiectasis due to other etiologies (assessing data specifically for each etiology), may be more informative and allow comparison of the effect of the etiology on the outcome.
Author Response
Response. We appreciate the reviewer’s helpful comments, which have substantially improved the quality of our study. As recommended, we compared the clinical characteristics of post-TB bronchiectasis with post-infectious, idiopathic, and other bronchiectasis in the revised manuscript. We have thoroughly modified the entire manuscript according to the reanalyzed results.
Reviewer 2 Report
General comment
The authors aim to describe the clinical characteristics of bronchiectasis patients with previous tuberculosis in Korea. As stated by the authors, Korea experienced a high incidence of tuberculosis in the 1960-1970 and this could explain the high incidence of TB in bronchiectasis patients included in the Korean registry. The structure of the KMBARC registry is similar to the EMBARC registry representing a strength of the study. Registries changed the landscape in bronchiectasis research and clinical practice leading to a tangible improvement in bronchiectasis care. Moreover, the authors well recognized some of the major limitations of the study.
Nonetheless, I do maintain a series of concerns.
Major comments
The authors stated that the attending physicians determined the aetiology of bronchiectasis based on the test results and questionnaires. Moreover, the attending physician determined pTB as the primary or as the secondary aetiology. However, physician judgment is highly subjective representing an important limitation. Although post-infective and previous tuberculosis bronchiectasis accounts for the majority of “known” causes of bronchiectasis in several cohorts (especially India and Africa), this definition suffers from several biases. There is consensus on neither the type and severity of previous infective processes which justify the development of bronchiectasis nor the acceptable time between them which should fit the “post-infective” definition (DOI: 10.1159/000455880). The post-infective bronchiectasis is a diagnosis of exclusion that should be hypothesized only when possible or definite diagnosis have been rule out. I think that this is a relevant limitation that should be discussed. Moreover, the aetiology of bronchiectasis depends on the intensity of the work-up performed (DOI: 10.1159/000455880). In the text is not specified if all the patients underwent the same “bundle” (according or not to the guidelines) to identify the cause of bronchiectasis. Thus, the group “other bronchiectasis” might be highly heterogenous including patients with PCD, ABPA, COPD, primary and secondary immunodeficiencies. For these reasons, a diagnostic objective algorithm has been developed by Aurujo et colleagues and validated in 10 bronchiectasis cohorts to identify the cause of bronchiectasis (DOI: 10.1183/13993003.01289-2017). I suggest to narrowing the group of other bronchiectasis patients to patients with idiopathic bronchiectasis in case all the patients underwent the same bundle of tests. In case not, this limitation should be discussed.
Page 4 line 154: I can’t see the clinical impact of the lower FEV1/FVC ratio in pTB patients compared to other bronchiectasis. Severity of airflow obstruction is defined by FEV1 whereas the presence of obstruction is defined by the FEV1/FVC ratio < than 0.7 (while some authors prefer to use the lower limits of normal to define airflow obstruction). I suggest to modify the sentence as following 1. the % of patients in each group with FEV1/FVC < 0.7 a 2. Severity of obstruction defined by FEV1 in each group.
Table 3: “The median value and interquartile range for the FACED score is the same for the two study groups although the p value is 0.019. Look for consistency.
Minor comments
Page 3 line 131: “The median patient age was 66 years (IQR 66-72 years)” but in table 1 the IQR is 60-72 years. Look for consistency.
Grammar and syntax should be carefully reviewed. For example Page 6 line 105: “The collected data comprise demographics, comorbidities; laboratory test including microbiology, radiology and pulmonary function findings; and treatment.”
Author Response
## Response to Reviewer 2
General comment. The authors aim to describe the clinical characteristics of bronchiectasis patients with previous tuberculosis in Korea. As stated by the authors, Korea experienced a high incidence of tuberculosis in the 1960-1970 and this could explain the high incidence of TB in bronchiectasis patients included in the Korean registry. The structure of the KMBARC registry is similar to the EMBARC registry representing a strength of the study. Registries changed the landscape in bronchiectasis research and clinical practice leading to a tangible improvement in bronchiectasis care. Moreover, the authors well recognized some of the major limitations of the study.
Nonetheless, I do maintain a series of concerns.
Response. We appreciate the reviewer’s helpful comments, which have substantially improved the quality of our study. We are submitting a revised manuscript that addresses these concerns. A detailed point-by-point response to these concerns is provided.
Major comments
Comment 1 (C1). The authors stated that the attending physicians determined the aetiology of bronchiectasis based on the test results and questionnaires. Moreover, the attending physician determined pTB as the primary or as the secondary aetiology. However, physician judgment is highly subjective representing an important limitation. Although post-infective and previous tuberculosis bronchiectasis accounts for the majority of “known” causes of bronchiectasis in several cohorts (especially India and Africa), this definition suffers from several biases. There is consensus on neither the type and severity of previous infective processes which justify the development of bronchiectasis nor the acceptable time between them which should fit the “post-infective” definition (DOI: 10.1159/000455880). The post-infective bronchiectasis is a diagnosis of exclusion that should be hypothesized only when possible or definite diagnosis have been rule out. I think that this is a relevant limitation that should be discussed. Moreover, the aetiology of bronchiectasis depends on the intensity of the work-up performed (DOI: 10.1159/000455880). In the text is not specified if all the patients underwent the same “bundle” (according or not to the guidelines) to identify the cause of bronchiectasis. Thus, the group “other bronchiectasis” might be highly heterogenous including patients with PCD, ABPA, COPD, primary and secondary immunodeficiencies. For these reasons, a diagnostic objective algorithm has been developed by Aurujo et colleagues and validated in 10 bronchiectasis cohorts to identify the cause of bronchiectasis (DOI: 10.1183/13993003.01289-2017). I suggest to narrowing the group of other bronchiectasis patients to patients with idiopathic bronchiectasis in case all the patients underwent the same bundle of tests. In case not, this limitation should be discussed.
Response 1 (R1). Thank you for pointing this out as we had not fully acknowledged it in our original manuscript. We agree with your concern that physicians’ judgement on the aetiology of bronchiectasis might be highly subjective. Thus, we have clearly mentioned this issue as a limitation of our study in the Discussion section of the revised manuscript (page 8, lines 265–270).
“Third, physicians’ judgments regarding the aetiology of bronchiectasis could be subjective. Although all patients enrolled in the KMBARC were evaluated using the same clinical report form (same tests and questionnaire), the intensity of the diagnostic workup might differ between institutions and physicians. Since a diagnostic bundle for bronchiectasis in South Korea has been developed (unpublished data), this issue should be addressed in future studies.”
C2. Page 4 line 154: I can’t see the clinical impact of the lower FEV1/FVC ratio in pTB patients compared to other bronchiectasis. Severity of airflow obstruction is defined by FEV1 whereas the presence of obstruction is defined by the FEV1/FVC ratio < than 0.7 (while some authors prefer to use the lower limits of normal to define airflow obstruction). I suggest to modify the sentence as following 1. the % of patients in each group with FEV1/FVC < 0.7 a 2. Severity of obstruction defined by FEV1 in each group.
R2. As recommended, we have modified the Results section of the revised manuscript (page 5, line 158).
“…the rate of patients with FEV1/FVC < 0.7 tended to be higher in patients with post-TB and post-infectious bronchiectasis than those with idiopathic and other bronchiectasis,…”
C3. Table 3: “The median value and interquartile range for the FACED score is the same for the two study groups although the p value is 0.019. Look for consistency.
R3. Thank you for your careful review of our manuscript. As you indicated, the difference in the FACED score between groups was not evident when the results were presented as median and interquartile range. Thus, we presented the FACED score as the mean and standard deviation (Table 3 on page 7).
Minor comments
C4. Page 3 line 131: “The median patient age was 66 years (IQR 66-72 years)” but in table 1 the IQR is 60-72 years. Look for consistency.
R4. Thank you for your careful review of our manuscript. We have corrected this in the Results section of the revised manuscript (page 3, line 131).
“The median patient age was 66 years (IQR, 60–72 years) and 56% were females.”
C5. Grammar and syntax should be carefully reviewed. For example Page 6 line 105: “The collected data comprise demographics, comorbidities; laboratory test including microbiology, radiology and pulmonary function findings; and treatment.”
R5. As recommended, the revised manuscript has been proofread by a professional native English speaker. The Methods section indicated by the reviewer was revised as follows (page 3, lines 88–91):
“The KMBARC registry collected baseline data at baseline (recruitment) and follow-up visits once a year. The collected data included demographics, comorbidities, laboratory test results (including microbiology), radiology, pulmonary function test results, and treatment”